# Description Logic Concept Learning using Large Language Models

**Adrita Barua**                    ADRITA@KSU.EDU  and  **Pascal Hitzler**                    HITZLER@KSU.EDU
*Kansas State University, Manhattan, KS, USA*

Editors: Leilani H. Gilpin, Eleonora Giunchiglia, Pascal Hitzler, and Emile van Krieken

## Abstract

Recent advances in Large Language Models (LLMs) have drawn interest in their capacity for logical reasoning, an area traditionally dominated by symbolic systems that rely on complete, manually curated knowledge bases represented in formal languages. This paper introduces a framework that leverages pretrained LLMs to generate Description Logic (DL) class expressions from instance-level examples and background knowledge, translated to natural language. The baseline is Concept Induction, a symbolic learning approach that is mostly based on formal logical reasoning over a DL theory. Drawing inspiration from the DL-Learner architecture, our approach replaces traditional symbolic methods with LLM-based models to generate DL class expressions from instance-level data.

We evaluate our approach using three benchmark ontologies across two LLMs: `gpt-4o` and `o3-mini`. We use a symbolic reasoner, Pellet, to verify the LLM-generated results and incorporate the reasoner's feedback into our pipeline to ensure logical consistency, thereby generating a hybrid neurosymbolic system. By introducing controlled variations to the background knowledge, we assess the models' reliance on commonsense versus formal reasoning. Results show that `o3-mini` achieves near-perfect accuracy across settings, albeit with longer runtime. These findings demonstrate that LLMs have the potential to serve as scalable and flexible DL learners when coupled in a hybrid neurosymbolic setting, offering a promising alternative to symbolic approaches—particularly in contexts where high-quality ontologies are incomplete or unavailable.

## 1. Introduction

Logical reasoning lies at the core of human cognition and artificial intelligence. Traditional approaches to logical reasoning in AI have relied heavily on formal languages for knowledge representation and symbolic reasoners for inference. However, these methods often suffer from brittleness and the well-known knowledge acquisition bottleneck, making them difficult to scale and apply in real-world scenarios (Musen and Van der Lei (1988)).

With the advent of Large Language Models (LLMs) (Brown et al. (2020); Touvron et al. (2023); Anil et al. (2023)), there has been increasing interest in evaluating their potential for reasoning. Several studies have investigated how these generative models perform in logical reasoning tasks (Xu et al. (2024b)). This paper proposes a framework that leverages pretrained LLMs to generate logical class expressions based on a semi-structured knowledge base and instance-level data, all represented in natural language. This approach addresses several limitations of traditional symbolic systems while offering advantages over end-to-end neural methods that lack interpretability.

We adopt the setting of Concept Induction, inspired by the DL-Learner framework (Bühmann et al. (2016)), where a background knowledge base in OWL is provided along

with sets of individuals categorized as positive and negative examples. The task is to derive a logical formula, typically an OWL *class expression*[1], that includes all or most of the positive examples and excludes all or most of the negative ones. In deriving its responses, DL-Learner invokes a formal logic reasoner many times, i.e. while Concept Induction is about the *learning* of class expressions from examples, it is at its core a symbolic reasoning task. In our case, instead of using a formal OWL ontology, we translate the background knowledge into a natural language format. This choice is motivated by the scarcity of high-quality ontologies, as building such ontologies requires significant resources.

We then use an LLM as the reasoning engine to learn from these positive and negative instances and to generate complex class expressions in *Manchester OWL Syntax*.[2] Using LLMs as logical reasoners in this setting, we aim to evaluate their effectiveness in deriving descriptive class expressions from examples. This investigation provides several major insights:

- This LLM-based method can automate different aspects of ontology engineering. While the number of knowledge bases in the Semantic Web continues to grow, the maintenance and creation of ontology schemata remains a challenge. In particular, creating class expressions constitutes one of the most demanding tasks in ontology engineering.

- This setting allows us to explore the potential of LLMs to perform deep deductive reasoning, a task that remains challenging for deep learning systems (Hitzler et al. (2025)). Deductive reasoning over formal logic remains a key challenge in symbolic AI (Brachman and Levesque (2004)), and enabling such reasoning in neural models opens new possibilities for neurosymbolic systems that combine learning and reasoning.

- An LLM-based Description Logic Learner can be used to generate human-understandable class expressions, offering interpretable concept definitions derived from instance data. This aligns with the goals of explainable AI (XAI), where such expressions can help explain the behavior of black-box models by linking input instances to their output classifications through logical descriptions (Dalal et al. (2024)).

Having an LLM-based Description Logic Learner can help alleviate the existing shortcomings of symbolic methods for concept learning. This method investigates the potential of generating an end-to-end automated system that can perform Concept Induction at scale, even when quality background knowledge is limited or unavailable, thus addressing a critical gap in existing methods.

The rest of the paper is organized as follows. In Section 2, we provide a brief overview of related research. In Section 3, we discuss the current DL-Learner framework and explain how we adapt it in our approach. Section 4 presents a detailed description of our methodology and outlines the steps in our pipeline, illustrating how LLM-based models are employed to generate Description Logic class expressions. In Section 5, we present and analyze the evaluation results. Finally, in Section 6, we conclude the paper and propose directions for future work.

---

1. https://www.w3.org/TR/owl2-syntax/#Class_Expressions
2. https://www.w3.org/TR/owl2-manchester-syntax/

## 2. Related Work

Initial efforts in symbolic reasoning methods relied heavily on Knowledge Representation and Reasoning (KR) systems, which, although foundational, proved difficult to scale and required considerable manual effort. Reasoning with formal languages has proven particularly challenging (Musen and Van der Lei (1988); Cropper et al. (2022)). Such systems tend to fail when complete knowledge is unavailable, suffer from the knowledge acquisition bottleneck, and require significant domain expertise for encoding knowledge in formal logic. These systems also struggle to handle raw, unstructured data such as natural language, are highly sensitive to labeling errors, and often fail to generalize across symbolically different but semantically similar concepts (Yang et al. (2023)).

Recent advances in Large Language Models (LLMs) offer promising alternatives. LLMs contain substantial implicit knowledge (Davison et al. (2019)), enabling them to handle incomplete information more robustly (Talmor et al. (2020)). They naturally process raw language data and can leverage massive web corpora to automatically construct rule bases (Ji (2018)). LLMs are also less sensitive to input errors (Meng et al. (2021)) and can recognize semantically similar concepts through embeddings (Mikolov et al. (2013)), making them well-suited for reasoning tasks in real-world settings.

Description Logics (DLs), which are, essentially, decidable fragments of first-order logic, are central to Semantic-Web-based knowledge representation and reasoning, in particular the Web Ontology Language OWL is based on them, see Hitzler et al. (2010). To automate the acquisition of OWL ontologies using machine learning, it is essential to develop tools capable of learning in description logics (Lehmann and Hitzler (2010)). Traditional symbolic learners face the limitations outlined above, and recent research shows that transformer-based models or LLMs have proven to be effective alternatives. LLMs have demonstrated strong performance across a variety of logical reasoning benchmarks (Liang et al. (2023); Srivastava et al. (2023); Fan et al. (2024); Wei et al. (2022a)), though it remains unclear whether they possess generalized logical reasoning abilities and to what extent they are capable of robust logical inference (Li et al. (2024b); Valmeekam et al. (2022)).

To address this gap, several recent studies have focused on enhancing the reasoning capabilities of LLMs (Li et al. (2024a); Morishita et al. (2024); Tong et al. (2024)). For instance, Hua et al. (2024) show that an LLM's performance can vary significantly depending on the context or domain of the task. Hong et al. (2024) evaluates the self-verification abilities of LLMs using a verification loop to improve LLM performance in logical reasoning problems. Techniques such as Chain-of-Thought (CoT) prompting (Wei et al. (2022b)) emulate human-like reasoning and have inspired further strategies such as Logic-of-Thought (Liu et al. (2025)), RATT (Zhang et al. (2025)), and others (Wang et al. (2023); Zhou et al. (2023); Li et al. (2023); Lightman et al. (2024)). These approaches aim to align LLM reasoning with human cognitive processes. Structured reasoning techniques like these have been applied in tasks involving logical inference and symbolic reasoning (e.g., Xu et al. (2025); Luo et al. (2023); Parmar et al. (2024)). They typically involve long chains of thought (Long CoT), incorporating reflection, backtracking, and validation. However, training LLMs for Long CoT reasoning is resource intensive, often relying on proprietary methods (Jaech et al. (2024)) or expensive to reproduce (Guo et al. (2025)). To mitigate

this, Li et al. (2025) propose a more efficient strategy using supervised fine-tuning (SFT) and parameter-efficient techniques like LoRA to train LLMs for Long CoT reasoning.

Some frameworks, such as Logic-LM (Pan et al. (2023)) and LINC (Olausson et al. (2023)), use LLMs as translators to convert natural language into symbolic representations, which are then processed by external reasoners. These approaches do not fully explore the internal reasoning capability of LLMs. To overcome this, Symbolic Chain-of-Thought (Xu et al. (2024a)) proposes a fully LLM-based reasoning framework that breaks tasks into modular substeps. Similarly, Cumulative Reasoning (CR) (Zhang et al. (2023)) adopts a collaborative multi-agent LLM framework that decomposes complex problems into subtasks and incrementally builds a solution by validating intermediate steps.

However, these approaches have not thoroughly investigated how LLMs perform in real-world reasoning scenarios, particularly when dealing with large sets of instance-level examples to produce inferred class expressions. Barua et al. (2024) discusses an LLM-based concept induction system in which the LLMs produce only simple concept descriptions (i.e., word sets that distinguish positive from negative examples). It this approach it only identifies distinguishing concepts rather than constructing a full logical definition and does not account for negations. In this work, we investigate whether an LLM can generate Description Logic class expressions from input instances, using DL-Learner (Bühmann et al. (2016)) as our baseline. We provide background knowledge in natural-language format to offer context and aim to achieve results comparable to DL-Learner.

Deng et al. (2024) demonstrated improvements in the reasoning capabilities of LLMs using an iterative feedback module, which passes unsuccessful execution results from an external solver back to the LLM, thereby improving its translation reliability and correctness. In our pipeline, we incorporate a similar fact checking mechanism using the Pellet reasoner, creating a feedback loop in which incorrect or invalid outputs from the LLM are passed back for refinement, enhancing output quality, and improving the overall reliability of the LLM.

## 3. DL-Learner framework

We use *DL-Learner*[3] as our baseline to replicate Description Logic (DL) class expressions using a Large Language Model (LLM). DL-Learner is a flexible framework designed for solving learning problems in OWL, offering a component-based architecture with support for various knowledge base formats, reasoner interfaces, and machine learning algorithms.

It consists of four main component types. For each type, multiple implementations are available, each with its own configurable options, as illustrated in Figure 1.

**Knowledge Sources** integrate background knowledge from standard OWL formats (e.g., RDF/XML, Turtle, Manchester OWL Syntax) and SPARQL endpoints. Multiple sources can be combined, allowing DL-Learner to scale to large and distributed knowledge bases.

**Reasoner Components** provide connections to both external and internal reasoners. DL-Learner offers DIG 1.1.5 and OWL API interfaces, enabling connections to standard OWL reasoners. Additionally, DL-Learner includes its own approximate reasoner, which

---

3. See official DL-Learner documentation at https://www.researchgate.net/publication/228877947_DL-Learner_Manual

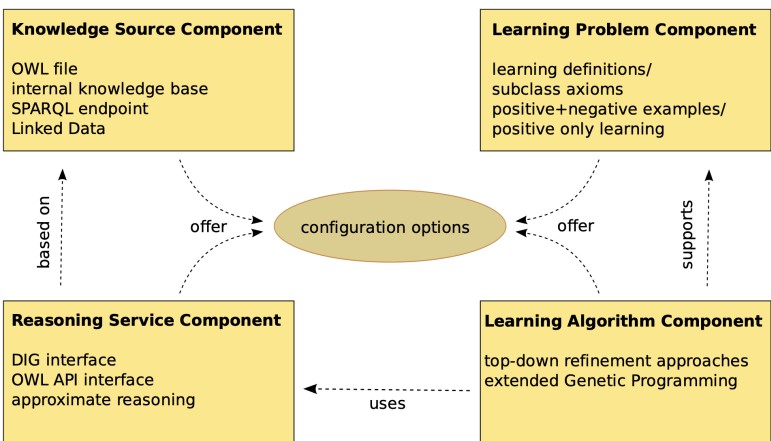

Figure 1: DL-Learner components

uses Pellet for bootstrapping and loading the inferred model into memory. Afterwards, instance checks are performed efficiently using a local closed world assumption (Badea and Nienhuys-Cheng (2000)).

**Learning Problems** specify the type of inference task to be solved, such as learning from positive and negative examples, positive-only learning, or class axiom learning. These components also incorporate efficient coverage check strategies for evaluating candidate expressions.

**Learning Algorithms** implement methods to solve the learning problems using a variety of strategies. Notably, refinement-operator-based algorithms iteratively construct more expressive class expressions through multiple refinement loops. In each loop, candidate concepts are extended or modified using syntactic transformations, guided by heuristics or stochastic methods to better fit the training examples. The framework includes algorithms based on genetic programming (Lehmann (2007)), refinement operators for $ALC$ (Lehmann and Hitzler (2007)), and extended operators that support OWL features and datatypes (e.g., Class Expression Learning for Ontology Engineering (CELOE), Refexamples (OCEL)), all optimized for generating concise, human-readable class expressions.

Together, these components enable DL-Learner to generate accurate and interpretable DL class expressions over complex and large-scale ontologies (for detailed description, see the official manual).[4] However, like other symbolic reasoners, it has certain limitations. It relies heavily on complete and high-quality background knowledge, which can cause failures when such knowledge is incomplete. It also cannot automatically recognize semantically similar concepts and does not scale well when the complexity of the background knowledge or the volume of instance examples increases. For example, in the top-down refinement approach, if a perfect solution does not exist, the algorithm may have very long run times—or run until it exhausts system memory.

---

4. https://www.researchgate.net/publication/228877947_DL-Learner_Manual

To overcome these challenges, we propose replacing the traditional learning algorithm with an LLM, which can analyze the learning problem based on the given instance examples and apply its own commonsense knowledge—along with the provided background information—to generate reasonable DL class expressions. This can break the infinite refinement loop and significantly improve scalability. Moreover, LLMs may be able to produce accurate and meaningful class expressions even when the background knowledge is incomplete or not available in a well-structured ontology format. Motivated by this, we developed a pipeline using LLMs to generate DL class expressions in a setting similar to that of DL-Learner.

## 4. Methodology

We create an end-to-end pipeline to generate DL class expressions using LLMs, as illustrated in Figure 2.[5]

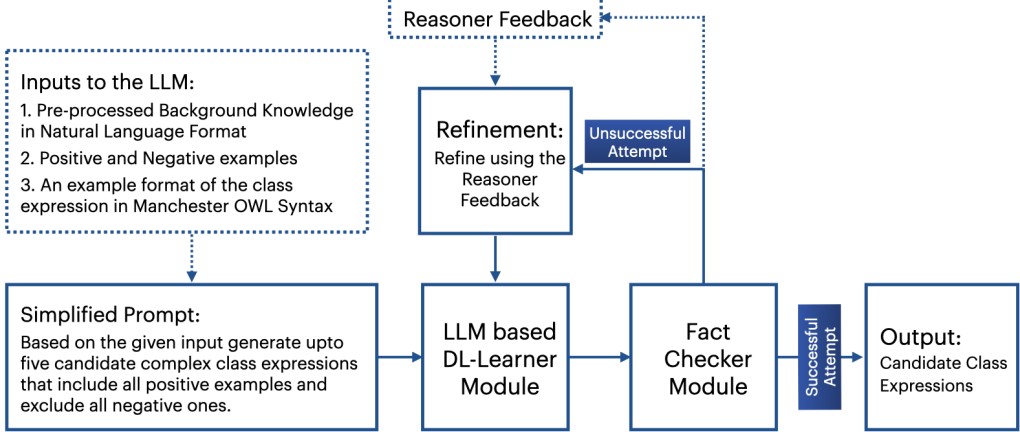

Figure 2: LLM-based DL-Learner framework. Key inputs to the LLM are: (1) background knowledge in natural language, (2) positive and negative examples, and (3) a sample class expression in Manchester OWL Syntax. These inputs construct a prompt that is passed to the LLM. Based on the prompt instructions, the LLM generates up to five candidate class expressions. The fact-checker module evaluates each candidate using a symbolic reasoner: if one is valid, it is returned; otherwise, feedback is sent back to the LLM for up to three refinement iterations. The full prompt is in Appendix A.

### 4.1. Preprocessing

The first step transforms the background ontology from OWL format into a natural language representation. We use a Python script built with the `rdflib` library[6] to extract classes, subclasses, individuals (instances of classes), object properties, and their domain/range information. This information is then expressed in natural language and stored in a `.txt` file. For example, if the ontology specifies that an individual `Anna` is an instance of the class

---

5. See https://github.com/AdritaBarua/DL-learner-using-LLMs for source code, input data, raw result files, and parameter settings for replication.

6. https://pypi.org/project/rdflib/4.0/

`Female`, and `Anna` has a `hasChild` relationship with another individual `Dino`, the script extracts and formats the following lines:

- 'Anna' is an instance of class 'Female'.
- 'Anna' has a relationship 'hasChild' with 'Dino'.

### 4.2. DL Class Generation Using LLMs

After converting the ontology to natural language, the content is saved in a `.txt` file and passed to the LLM as background knowledge.

We use two GPT models: `gpt-4o` (Hurst et al. (2024)) and `o3-mini`,[7] invoked via the OpenAI API. `gpt-4o` is OpenAI's flagship model optimized for speed and cost, making it suitable for simple reasoning tasks. In contrast, `o3-mini` is a reasoning model trained using reinforcement learning to solve more complex problems. It generates internal chains of thought before responding, offering better reasoning capabilities at the cost of slower performance. For `gpt-4o`, we set the `temperature` to 0 and `top_p` to 1. For `o3-mini`, we set the reasoning effort to `"high"`. Both models are prompted with the system message: `"You are a helpful Description Logic learner."`

We use three small to moderately sized benchmark ontologies: `Trains`, `Basic Family`, and `Family Benchmark` (Lehmann and Hitzler (2010)). After preprocessing, these ontologies are passed to the LLMs as background knowledge. We employ few-shot prompting, where each prompt includes: (1) background knowledge in natural language, (2) positive and negative examples, and (3) a sample class expression in Manchester OWL Syntax.

Given these inputs, the LLM is instructed to generate a complex class expression that includes all positive examples and excludes all negative ones. For instance, to define the concept `Brother`, the LLM may generate: `Male and (hasSibling some Thing)`. This expression is inferred from the background knowledge, where positive examples are instances of `Male` with `hasSibling` relations, while negative examples are either `Female` or don't have sibling relationships. We provide a step-by-step prompt structure, as shown in Appendix A, and instruct the LLM to return up to five candidate class expressions of minimal length.

### 4.3. Fact-Checking Using a Reasoner

To validate the class expressions generated by the LLM, we use a symbolic reasoner. Specifically, we use the same reasoner component integrated into the DL-Learner framework (Bühmann et al. (2016)). This module is implemented in Java and uses Pellet as the base reasoner. Pellet loads the inferred model into memory and performs instance checks under a closed-world assumption. Each of the five candidate expressions is evaluated for correctness using this reasoner.

### 4.4. Refinement Using a Feedback Loop

If at least one candidate expression is found to be 100% accurate—i.e., it correctly includes all positive examples and excludes all negative ones—the process terminates, and the result is saved. If none of the expressions are valid or sufficiently accurate, feedback from the reasoner is appended to the original prompt and sent back to the LLM for refinement. This

---

7. https://openai.com/index/o3-o4-mini-system-card/

loop continues for up to three iterations or until a valid class expression is found. If all the attempts are failed, the result is noted as invalid. Appendix B illustrates an example of the feedback given to the LLM after an unsuccessful attempt.

## 5. Evaluation Results

Two variations of the prompt were used to verify reasoning in different contexts, to check if the LLM models perform commonsense reasoning when obvious interpretations exist— thereby generating the DL class expressions without requiring logical inference from the input data. We evaluated three ontologies (Trains, Family Benchmark, and Basic Family) under two core prompt-engineering conditions for both GPT models (gpt-4o and o3-mini):

- **With basename prompt:** "Give me the complex class expression for the relation (e.g., `Brother`) based on the given examples."
- **Without basename prompt:** "Give me the complex class expression based on the given examples."

For the Basic Family ontology, we also tested two additional variations:

- **Changed gender:** The underlying ontology flips the genders of all individuals while keeping the relation names unchanged. Thus, when the prompt asks to generate the class expression for "Brother," it should produce the expression for "Sister" based on the modified ontology.
- **Changed relations:** The ontology remains the same, but the prompt intentionally renames the target relation to a misleading term.

For the Trains ontology, no simple concept name is available to use as a commonsense hint, so only the "without basename prompt" was used in that case. All outputs were then verified using the same reasoner module employed by DL-Learner, ensuring consistency with DL-Learner's results. All the results are shown in Table 1. Here, the success rate indicates the ratio of logically valid results produced by the LLMs across the sample set, as determined by the reasoner's feedback (see Section 4.4). The DL learner achieves a 100% success rate on these benchmark ontologies.

### 5.1. Discussion

Including the relation name in the prompt ("Basename=Yes") generally improves accuracy for both models. The general model (gpt-4o) fails in most cases when the relation name is omitted, indicating that it relies heavily on commonsense cues rather than pure logical inference (Figure 3). The failure rate is particularly high for the Family Benchmark dataset, where the ontology and example sets are larger. The reasoning model (o3-mini) achieves an almost perfect success rate regardless of whether the basename is provided (Figure 4). However, its processing time is high. It mainly fails on a few cases in the Family Benchmark, suggesting that o3-mini also struggles when ontology complexity and example-set size increase. Interestingly, when a wrong relation name is given (the "changed_relation" prompt) to confuse the model, o3-mini still succeeds. Although this misleading relation name should confuse gpt-4o, its success rate in that scenario remains higher than when using the default ontology without a basename hint. The general model (gpt-4o) also struggles with the Trains ontology, despite that the ontology and its example sets being smaller than those in the Family Benchmark.

Table 1: Success rate of gpt-4o and o3-mini across different ontology datasets and prompt variations. The success rate is the ratio of logically valid results produced by the LLM across the sample set.

| Sample size | Dataset/Ontology | Variation | Model | Basename Prompt | Success Rate (%) |
|---|---|---|---|---|---|
| 9 | basic_family.owl | default | gpt-4o | No | 67 |
| 9 | basic_family.owl | default | gpt-4o | Yes | 100 |
| 9 | basic_family.owl | default | o3-mini | No | 100 |
| 9 | basic_family.owl | default | o3-mini | Yes | 100 |
| 17 | family_benchmark.owl | default | gpt-4o | No | 18 |
| 17 | family_benchmark.owl | default | gpt-4o | Yes | 59 |
| 17 | family_benchmark.owl | default | o3-mini | No | 59 |
| 17 | family_benchmark.owl | default | o3-mini | Yes | 94 |
| 5 | trains.owl | default | gpt-4o | No | 40 |
| 5 | trains.owl | default | o3-mini | No | 100 |
| 9 | basic_family.owl | changed_gender | gpt-4o | No | 56 |
| 9 | basic_family.owl | changed_gender | gpt-4o | Yes | 78 |
| 9 | basic_family.owl | changed_gender | o3-mini | No | 100 |
| 9 | basic_family.owl | changed_gender | o3-mini | Yes | 100 |
| 9 | basic_family.owl | changed_relations | gpt-4o | Yes | 78 |
| 9 | basic_family.owl | changed_relations | o3-mini | Yes | 100 |

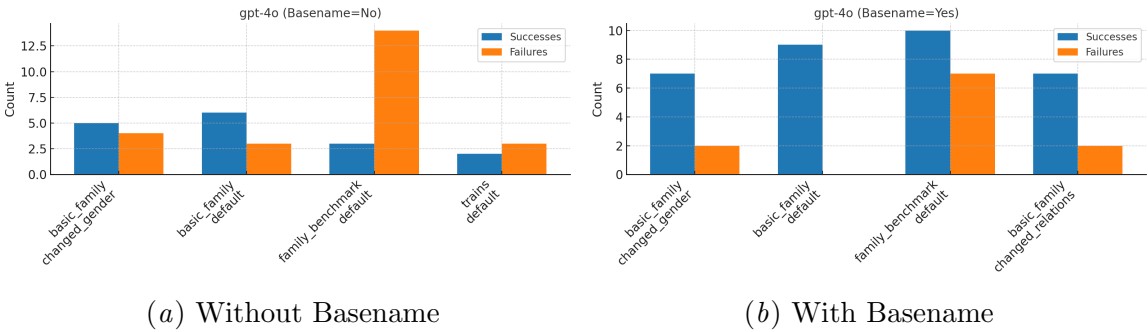

(a) Without Basename          (b) With Basename

Figure 3: Success rate of gpt-4o

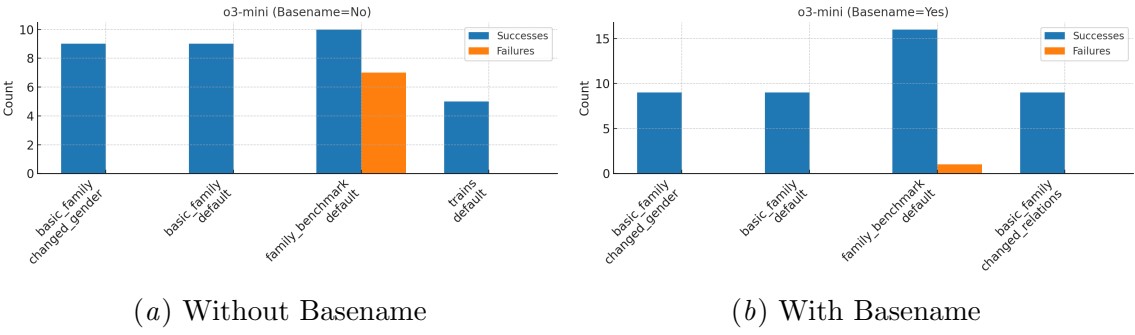

(a) Without Basename          (b) With Basename

Figure 4: Success rate of o3-mini

### 5.2. Qualitative Analysis

We evaluate success and failure rates using the reasoner; however, the output class expressions do not exactly match DL-Learner's, since multiple valid class expressions can satisfy the same set of examples. DL-Learner's learning component finds the *shortest* valid class expression, whereas the LLM—despite being instructed to produce the shortest expression—does not always do so (see Table 2). In some cases, the LLM reasoning model generates a valid expression by trivially excluding all negative examples, rather than finding the minimal generalization that covers all positives and excludes negatives (e.g., in the Family Benchmark ontology). This suggests the prompt does not explicitly force the model to find a truly generalized class expression. Future work should refine the system to improve the quality of LLM-generated expressions.

Table 2: Average shortest class expression lengths for DL-Learner and o3-mini (without basename prompt) across three datasets in their default setting

| Dataset | DL-Learner Avg Length | o3-mini Avg Length |
|---|---|---|
| trains.owl | 3 | 5 |
| basic_family.owl | 4 | 5 |
| family_benchmark.owl | 7 | 10 |

## 6. Conclusions and Future Work

This paper discusses how LLMs can be used to learn DL class expressions from instance examples. We introduced an LLM-based DL-Learner framework in which the model processes background knowledge in natural-language format, mitigating reliance on formal knowledge bases for description logic reasoning. We evaluated whether the models rely on common-sense cues, pure logical inference, or a mixture of both. The results indicate that, even though the reasoning model (o3-mini) achieves near-perfect accuracy for most dataset variations, the quality of the generated class expressions can still be improved. However, these findings demonstrate the potential of using LLMs to learn logical class expressions from example data points when high-quality background knowledge is not available.

Our goal here is to leverage LLMs in a real-world scenario so that they can function as standalone tools for ontology engineering, data learning, and certain XAI tasks at scale with high-quality output. Further modifications—using fine-tuning and reinforcement learning methods—can produce an agentic system capable of reliable, large-scale description logic learning.

### Acknowledgments

The authors acknowledge partial funding under the National Science Foundation grant 2333782 "Proto-OKN Theme 1: Safe Agricultural Products and Water Graph (SAWGraph): An OKN to Monitor and Trace PFAS and Other Contaminants in the Nation's Food and Water Systems."

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

## Appendix A. First Appendix

The few shot prompt for generating DL class expressions is shown in Figure 5. This is the "with basename" prompt for the Basic Family ontology, where the name of the relation is explicitly mentioned. The line mentioning the basename is omitted in the "without basename" prompt.

## Appendix B. Second Appendix

An example of the feedback from the reasoner after an unsuccessful attempt, fed to the LLM for refinement, is shown in Figure 6.

Take the given ontology as the knowledge base: {kb_text}.

Use this knowledge base to extract complex class expressions for the given positive and negative examples. A complex class expression is an OWL class expression that consists of the classes and properties that apply to the positive examples but not to the negative examples.
Use Manchester syntax for the complex class expressions.
For example:
If the positive examples are:
(Dino, Luigi, Mauro, Francesco, Giuseppe)

And the negative examples are:
(Antonella, Giovanna, Maria, Marisella)

Then the complex class expression generated by DL-Learner is (first 5 are shown):
1: Male and (hasSibling some Thing)
(accuracy 100%, length 5, depth 1)
2: Male and ((not (Male)) or (hasSibling some Thing))
(accuracy 100%, length 8, depth 1)
3: Male and ((hasChild some Thing) or (hasSibling some Thing))
(accuracy 100%, length 9, depth 1)
4: Male and ((hasSibling some Thing) or (hasSibling some Thing))
(accuracy 100%, length 9, depth 1)
5: Male and ((hasSibling some Thing) or (hasParent max 1 Thing))
(accuracy 100%, length 10, depth 1)

All the class expressions are generated using the Closed World Assumption of the knowledge base.
Now, perform a logical reasoning based on the knowledge base to find the shortest complex class expressions for the following examples:

Positive examples: {pos_arg}
Negative examples: {neg_arg}
Generate the complex class expression for {base}. [Omit this line for "without basename" prompt]
{feedback}
Don't give me any description just the shortest complex class expressions (upto first 5) and follow the example format:

Figure 5: Prompt

```
Reasoner feedback:
Previous attempt:
1: hasCar some (OpenCar and ShortCar)
2: Train and (hasCar some (OpenCar and ShortCar))
3: (not (ClosedCar)) and (hasCar some (OpenCar and ShortCar))
Reasoner output:
Reasoner: Pellet
—— Expr 1 ——
LLM expr: hasCar some (OpenCar and ShortCar)
Positive ex. east2 —> true
Positive ex. east3 —> true
Positive ex. east1 —> true
Negative ex. west9 —> true
Negative ex. west7 —> true
Reasoner accuracy for expr 1: 60.00% (3/5 correct)
Expr 1 is INVALID.
—— Expr 2 ——
LLM expr: Train and (hasCar some (OpenCar and ShortCar))
Positive ex. east2 —> true
Positive ex. east3 —> true
Positive ex. east1 —> true
Negative ex. west9 —> true
Negative ex. west7 —> true
Reasoner accuracy for expr 2: 60.00% (3/5 correct)
Expr 2 is INVALID.
—— Expr 3 ——
LLM expr: (not (ClosedCar)) and (hasCar some (OpenCar and ShortCar))
Positive ex. east2 —> true
Positive ex. east3 —> true
Positive ex. east1 —> true
Negative ex. west9 —> true
Negative ex. west7 —> true
Reasoner accuracy for expr 3: 60.00% (3/5 correct)
Expr 3 is INVALID.
=== Overall accuracy: 0.00%
Please refine based on the Reasoner feedback.
```

Figure 6: Feedback from the reasoner

