# OpenReview forum: "Description Logic Concept Learning using Large Language Models"
_nesyconf.org/NeSy/2025/Conference_Phase_2 — NeSy 2025 - Phase 2 Poster_

### Official Review · Reviewer_b5ZE · 2025-07-06
**review of Submission48**

**Rating:** 7
**Confidence:** 4

**Review:**

## summary of the paper

This paper deals with the topic of 'concept learning' --- a notion familiar in the Description Logic and Semantic Web communities where it pertains to the learning of OWL class descriptions (aka Description Logic concepts) from example individuals. The research examines the ability of pretrained LLMs to generate OWL class expressions (in Manchester OWL syntax), given (extensive) promptings consisting of: 1) background knowledge in the form of an OWL ontology converted into natural language, 2) sets of example individual class members (both positive and negative), 3) an example scenario consisting of positive and negative individuals and multiple corresponding valid OWL class expressions in Manchester OWL syntax, and 4) instructions to generate up to 5 OWL class expressions, in Manchester OWL syntax, that capture the positive example individuals but not the negative ones, per (2).

The LLM (candidate) outputs (OWL class expressions, in Manchester OWL syntax) are vetted before being released as offical outputs. The vetting involves using an OWL reasoner (Pellet) to evaluate each OWL class expression (to verify its syntactic validity and to check how well it includes the positive individuals and excludes the negative individuals). If none of the outputs are valid or sufficiently accurate, feedback from the reasoner is appended to the original prompt and the augmented prompt is fed back into the LLM for iterative refinement of the outputs.

The paper describes and discusses the use of a symbolic concept learning system called DL-Learner (developed in 2007 and reintroduced in a 2016 paper), and the paper purports to use DL-Learner as a baseline system against which to compare the performance of the LLM-based concept learning systems. The predictive performance results of the baseline system, DL-Learner, are not reported or discussed, however.

The research uses two OpenAI LLMs: gpt-4o and 03-mini. They are shown to have high success rates at generating correct (or good) OWL class expressions.  The o3-mini LLM, which is considered a 'reasoning LLM model', is shown to perform at 100% success rate most of the time, but at the cost of slower response times. The LLM gpt-4o achieves 100% success rate in several of the experiment scenarios.


## evaluation of the paper

**clarity / use of language / quality of writing**

The paper is clear, well written, and pleasant to read.  In places it feels a bit repetitive --- re-expressing things that have already been established in the reader's mind.

**novelty / originality**

The paper's novelty / orginality is modest.

Several aspects of the research are not novel:
* the authors reuse an experimental setup established by the DL-Learner symbolic framework, and they reuse DL-Learner itself as their baseline symbolic concept learning system
* a related-work paper they cite studies their topic (concept learning using LLMs), but using a different and simpler learning task
* it is known that LLMs are good at code generation, and their ontology generation capabilities have been studied as well
* LLM prompting strategies for tuning and enhancing the performance of pretrained LLMs have been and are being studied extensively in diverse contexts and application tasks.

The feature of the research that strikes me as most novel is the iterative refinement scheme they build around the LLM --- especially the fact that the LLM candidate outputs are assessed using symbolic reasoning, i.e. the symbolic reasoning services of the DL-Learner framework (which uses the OWL reasoner Pellet).  This demonstrates an attractive neurosymbolic feedback loop, where proper formal, symbolic, logical inference is used to refine subsymbolic LLM performance.

**impact / significance**

The paper's impact / significance is modest.

The main finding is that, given sufficiently extensive prompts, LLMs can do a good job at generating correct OWL class expressions for example sets of individuals. But LLM research of this type is a crowded field, and many similar tasks have been studied and similar results reported. This research largely reconfirms general LLM capabilities that are already fairly well established.

**other observations**

I think the claims about and references to LLM reasoning capabilities are over-cooked. An LLM is not a *reasoning engine*, and the high success rates reported for the LLMs do not equate to evidence of LLM reasoning capability, even though (to humans) the task at hand would appear to require reasoning to solve. The LLM outputs may well have been (and probably are) generated via means that have little to do with reasoning.

I was happy to learn about DL-Learner in Section 3, but I'm not sure this much detail about it really belongs in the paper.  We don't (and shouldn't) need to understand the inner workings of DL-Learner to understand the LLM-based research being described in the paper.

The performance metric 'success rate' is never defined. At one point the authors mention that success rates are evaluated by the reasoner (in DL-Learner), but they don't explain what these success rates represent or measure.  The absence of a definition for the central performance metric hurts the paper's scientific credibility.

The predictive performance results that are reported mention only scores for the two LLMs, not for the baseline symbolic system, DL-Learner. The reader has no opportunity to compare LLM performance with baseline performance. And all of the description and discussion of DL-Learner throughout the paper is wasted.  The absence of baseline performance scores hurts the paper's scientific credibility.

**Anonymity:**

Remain anonymous

---

### Official Review · Reviewer_fCbW · 2025-07-07
**This paper explores the use of LLMs for learning logical concepts. It proposes a Description Logic (DL) framework where LLMs generate candidate class expressions for logic reasoning tasks, and these expressions are then validated using a symbolic reasoner to ensure formal correctness.**

**Rating:** 5
**Confidence:** 4

**Review:**

This paper explores the use of LLMs for learning logical concepts. It proposes a Description Logic (DL) framework where LLMs generate candidate class expressions for logic reasoning tasks, and these expressions are then validated using a symbolic reasoner to ensure formal correctness.
My main doubt after reading the paper is that, in the end, Pellet – the critical component of the symbolic reasoner – still plays a central role in this architecture. Although the LLM generates candidate class expressions, Pellet serves as the ultimate evaluator. Without Pellet, there is no way to verify whether these expressions are logically valid with respect to the ontology and example instances under the current architecture, which only supports one-way validation. In other words, the LLM alone does not have grounded formal reasoning capabilities; it is Pellet that ensures the outputs satisfy formal DL semantics. Therefore, the authors cannot claim that “these findings demonstrate that LLMs can serve as scalable and flexible DL learners, offering a promising alternative to symbolic approaches,” because their approach still fundamentally relies on symbolic reasoning for correctness.

**Anonymity:**

Remain anonymous

---

### Official Review · Reviewer_FUuJ · 2025-07-09
**Interesting experiment**

**Rating:** 6
**Confidence:** 3

**Review:**

This paper presents an interesting investigation into the logical capabilities of LLMs. The specific task of DL concept induction is evaluated in a complex setup combining LLMs and a symbolic reasoner. The results show that a GPT4 o3 can perform this reasoning task with high success rates.

This paper makes an interesting contribution in combining LLMs with symbolic reasoning to perform a reasoning task. The results show that for small problems, the LLM (using also the symbolic reasoner) can perform the task effectively. There are some failures, though. It would be interesting to study more systematically how the performance changes when the problem complexity increases. Also the relationship between problem complexity and computation cost would be interesting to explore.

Minor points:
Section 2 "passive intermediaries rather than as active reasoner" I don't see why translation into formal logic is 'passive' and reasoning 'active'. I would suggest a different wording.

**Anonymity:**

Remain anonymous